# A numerical study of bounds in the correlations of fractional quantum Hall states

**Prashant Kumar and Frederick Duncan Michael Haldane**

Department of Physics, Princeton University, Princeton NJ 08544, USA

## Abstract

We numerically compute the guiding center static structure factor $\bar{S}(k)$ of various fractional quantum Hall (FQH) states to $\mathcal{O}\left((k\ell)^6\right)$ where $k$ is the wavenumber and $\ell$ is the magnetic length. Employing density matrix renormalization group on an infinite cylinder of circumference $L_y$, we study the two-dimensional limit using $L_y/\xi \gg 1$, where $\xi$ is the correlation length. The main findings of our work are: 1) the ground states that deviate away from the ideal conformal block wavefunctions, do not saturate the Haldane bound, and 2) the coefficient of $O\left((k\ell)^6\right)$ term appears to be bounded above by a value predicted by field theories proposed in the literature. The first finding implies that the graviton mode is not maximally chiral for experimentally relevant FQH states.



# 1  Introduction

The fractional quantum Hall (FQH) states are some of the best studied examples of topological phases of matter. They are characterized by various topological quantities such as quasiparticle charge, Hall conductance, Hall viscosity, and chiral central charge of the edge theory, that fundamentally arise from nontrivial correlations between electrons. A particularly useful measure of correlations in these states is the "guiding center" static structure factor $\bar{S}(\boldsymbol{k})$, which is quartic in wavenumber at long wavelengths in the presence of translation and inversion symmetries [1, 2]. A fundamental feature of the FQH ground states is that the fourth rank tensor that determines this quartic term satisfies the so called "Haldane bound" [2, 3], a lower bound on the strength of long-wavelength density fluctuations in terms of the Hall viscosity tensor [4, 5]. In the rotationally invariant case, when both the quartic term in the guiding center static structure factor and the Hall viscosity tensor are determined by one parameter each, the bound can be expressed as a simple scalar inequality between the two. At a physical level, it can be understood as the presence of a minimal amount of correlations that distinguish QH states from topologically trivial product states, i.e., the former cannot be adiabatically deformed to the latter.

Much work on the FQH has involved a class of rotationally-invariant model wavefunctions (Laughlin [6], Moore-Read [7], Read-Rezayi [8]) that are related to Euclidean conformal field theory, and saturate the Haldane bound [9, 10]. These model states are the highest-density states that belong to the kernel (zero-energy eigenstates) of certain very special model Hamiltonians, and have played a key role in understanding the FQHE. One of their very special features is that they are "maximally chiral", in that their entanglement spectrum in cylindrical geometry only contains contributions of one chirality relative to the the semi-infinite state with all particles on one side of the cut. This is a very strong condition for "maximal chirality": A weaker version of maximal chirality is that the low-lying part of the entanglement spectrum (or, equivalently, the topological edge modes) only has contributions of one chirality. This weaker version is usually satisfied by the ground states of Hamiltonians that perturb away from the model ones.

In this paper, we address the question - What conditions are required to saturate the Haldane bound? We show in Appendix B that continuous rotational invariance is a requirement. This is so because the fluctuations of angular momentum contribute to the static structure factor at $\mathcal{O}\big((k\ell)^4\big)$ but not to the Hall viscosity tensor. For rotationally invariant systems, it has been shown previously [11–13] that $\nu_- = p/(2np-1)$ Jain states [14] for $n > 1$, that do not satisfy the weak maximal chirality condition, do not saturate the bound either. These FQH states contain both chiralities of spin-2 graviton excitations above the rotationally invariant ground states. This then leads to a larger amount of correlations in the static structure factor at long-wavelengths than necessitated by the magnitude of Hall viscosity. However, it has been unclear whether one requires the strong maximal chirality or the weaker version is sufficient to saturate the bound for isotropic FQH states. In particular, some studies have supported the latter [9].

We investigate this question numerically and provide definitive evidence that weak maximal chirality is insufficient. Thus, we expect only the ideal conformal block wavefunctions to saturate Haldane bound. We compute the static structure factors in the long wavelength limit for FQH states at $\nu = 1/3, 1/5,$ and $2/5$ using rotationally invariant two-dimensional Hamiltonians. To this end, we use density matrix renormalization group on an infinite cylinder [15] of circumference $L_y$ and approach the 2D-limit by considering large $L_y/\ell$. We compute $\bar{S}_4$, the coefficient of $\mathcal{O}\big((k\ell)^4\big)$ term in the long-wavelength expansion of guiding center static structure factor, and show that it is larger than the Haldane bound for FQH ground states of generic Hamiltonians. We substantiate this observation by analyzing perturbations around the model

Hamiltonians of Laughlin FQH states at $\nu = 1/3$ and $\nu = 1/5$. It is shown that $\bar{S}_4$ increases above the bound as the perturbation is introduced.

An important ancillary result is the first numerical computation of $\bar{S}_6$, the coefficient of the $\mathcal{O}\left((k\ell)^6\right)$ term in the expansion of guiding center static structure factor. Previous studies based on field theories have conjectured an expression for $\bar{S}_6$ that involves the chiral central charge of edge theory [16–20]. We provide evidence for the conjecture using the matrix-product state representation of the Moore-Read wavefunction [21, 22] and show agreement with Ref. [23]. Similar to $\bar{S}_4$, the conjectured field theory expression of $\bar{S}_6$ is found to not be a strict equality in generic FQH states. We make an interesting empirical observation that analogous to the Haldane bound, the field theory prediction could be interpreted as an *upper* bound, although a rigorous inequality relating $\bar{S}_6$ to some analog of the Hall viscosity has not been found so far. An analysis of perturbations around the model Hamiltonians at $\nu = 1/3$ and $\nu = 1/5$ provides persuasive numerical evidence for this conjecture.

Here we utilize the composite-boson formulation to express the topological properties associated with fractional quantum Hall ground states. One of the authors of this paper has recently proposed [24] that FQH states are electric quadrupole fluids that arise due to the spontaneous formation of composite-bosons, the underlying quadrupole. This is reflected in the fact that it determines fluid dynamical properties such as the Hall viscosity tensor, even when continuous rotational symmetry is absent. Moreover, Chern-Simons level corresponding to the composite-boson can be used to infer quasi-particle charge and Hall conductivity. The composite-boson theory has the advantage of naturally separating the guiding center and Landau orbit degrees of freedom, thus distinguishing intra-LL correlations from the inter-LL ones, which becomes apparent when particle-hole transformations are analyzed. In the interest of making the paper self-contained, we review this formulation in section 3.

This paper is organized as follows. In section 2, we review some general properties of the guiding center static structure factor that measures only the intra-LL correlations without assuming rotational invariance. A review of the composite-boson formulation, that is used to express the topological quantities, is presented in section 3. The derivation and statement of the Haldane bound with only translational and inversion symmetries can be found in Ref. [2] and is reproduced in appendix A. It is shown in appendix B that rotational invariance is required for the saturation. Therefore, we specialize to the isotropic FQH states starting in section 4. Our numerical results, showing that neither the Haldane bound is saturated nor the conjectured expression of $\bar{S}_6$ is universal for rotationally invariant but weakly maximally chiral fractional quantum Hall ground states, are presented in section 5. This conclusion is further substantiated through an analysis of perturbations around ideal Hamiltonians in section 6. Our findings are summarized alongside remarks on experimental relevance in section 7. Lastly, we interpret two of the spectral sum rules derived in Ref. [25] in a purely lowest Landau level formulation in appendix C.

## 2 Guiding center static structure factor

In this section, we review some general properties of the static structure factor of FQH states in the presence of translation and inversion symmetries, without invoking rotational invariance a priori. In appendix B, we show that the saturation of Haldane bound requires rotational invariance, therefore we will specialize to only the isotropic case for numerical investigations. Nevertheless, the following two sections are formulated in a more general setting since it is not a requirement for the existence of FQH states or the Haldane bound, and experiments break continuous rotational symmetry. Moreover, relaxing this condition has previously led to the discovery of geometric fluctuations as a fundamental degree of freedom in FQH states [26].

We will here adopt a slightly different normalization of the guiding-center structure factor which is better adapted to the quantum Hall physics, in which it is normalized by the density of magnetic flux quanta. This is in contrast to the usual normalization by the particle density, which is standard in the theory of one-component uniform fluids, where the conventional structure factor is given by

$$S(\boldsymbol{k}) = \lim_{N\to\infty} \frac{1}{N} C(\rho(\boldsymbol{k}), \rho(-\boldsymbol{k})),$$ (1)

$$\rho(\boldsymbol{k}) = \sum_{i=1}^{N} e^{-i\boldsymbol{k}\cdot\boldsymbol{x}_i},$$ (2)

where $N$ is the number of particles, $C(A,B) = C(B,A) = \langle \frac{1}{2}\{A,B\}\rangle - \langle A\rangle\langle B\rangle$ is the correlator of two operators (which in general do not commute, although they do in this case), and $C(A^\dagger, A)$ is real, as in (1), where $S(-\boldsymbol{k}) = S(\boldsymbol{k})$. In a uniform magnetic field, the Faraday tensor in the 2D plane is parameterized by

$$F_{ab} = \partial_a A_b(\boldsymbol{x}) - \partial_b A_a(\boldsymbol{x}).$$ (3)

This can be written as $F_{ab} = B\epsilon_{ab}$, where $\epsilon_{ab}$ is the 2D antisymmetric (Levi-Civita) symbol, which has a sign ambiguity. The usual convention in the FQH field is to chose the sign of $\epsilon_{ab}$ so that $eB > 0$, but it is preferable to use quantities like $F_{ab}$ that are independent of the sign (handedness) convention. The electron coordinates can be written as the sum of a guiding center coordinate $\bar{\boldsymbol{R}}_i$ plus a Landau orbit radius vector $\tilde{\boldsymbol{R}}_i$

$$\boldsymbol{x}_i = \bar{\boldsymbol{R}}_i + \tilde{\boldsymbol{R}}_i,$$ (4)

which have the commutation relations

$$[\bar{R}_i^a, \bar{R}_i^b] = [\tilde{R}_i^b, \tilde{R}_i^a] = -i\frac{\hbar}{eB}\epsilon^{ab}, \quad [\bar{R}_i^a, \tilde{R}_i^b] = 0,$$ (5)

which preserves the relation $[x_i^a, x_i^b] = 0$. The decomposition of $\boldsymbol{x}$ into guiding center plus Landau orbit-vector has an ambiguity $(\bar{\boldsymbol{R}}_i, \tilde{\boldsymbol{R}}) \to (\bar{\boldsymbol{R}}_i + \boldsymbol{a}, \tilde{\boldsymbol{R}}_i - \boldsymbol{a})$. The closed semiclassical Landau orbit $\boldsymbol{k}_n(t)$ in the Brillouin zone has a period-averaged mean value $\boldsymbol{k}_n^0$, and the orbit-vector is $\hbar(k_n(t) - k_n^0)_a = eF_{ab}\tilde{R}^b$, so all intra-Landau-level matrix elements $\langle n\alpha|\tilde{\boldsymbol{R}}|n\beta\rangle$ are chosen to vanish, where $n$ is the Landau level index and $\alpha, \beta$ are states within the Landau level. This can be done consistently when the number of particles in each Landau level is fixed. This means that a Landau orbit has no mean electric dipole moment relative to the guiding center. The total electric dipole moment operator is then given in terms of the guiding centers by $\boldsymbol{d} = e\sum_i \bar{\boldsymbol{R}}_i$, and the generator of translations (momentum operator) has components $-F_{ba}d^a$. (These formulas distinguish upper ($x^a$) and lower ($\partial_a \equiv \partial/\partial x^a$) spatial indices, which is necessary unless there is a continuous rotational symmetry defined by a metric.)

The guiding-center structure factor $\bar{S}(\boldsymbol{k})$ in a partially-occupied Landau level with filling $\nu$, $0 < \nu < 1$ will be defined here by

$$\bar{S}(\boldsymbol{k}) = \lim_{N\to\infty} \frac{\nu}{N} C(\bar{\rho}(\boldsymbol{k}), \bar{\rho}(-\boldsymbol{k})),$$ (6)

$$\bar{\rho}(\boldsymbol{k}) = \sum_{i=1}^{N} e^{-i\boldsymbol{k}\cdot\bar{\boldsymbol{R}}_i},$$ (7)

where the sum over $N$ particles is restricted to the ones in the partially-occupied Landau level, and $\bar{\rho}(\boldsymbol{k})$ obeys the Girvin-Macdonald-Platzman (GMP) Lie algebra

$$[\bar{\rho}(\boldsymbol{k}), \bar{\rho}(\boldsymbol{k}')] = 2i\sin\left(\tfrac{\hbar}{2eB}\epsilon^{ab}k_a k_b\right)\bar{\rho}(\boldsymbol{k} + \boldsymbol{k}').$$ (8)

The virtue of the improved definition (6) is that $v/N = 1/N_\Phi$, where $N_\Phi$ is the number of London flux quanta passing through the system, and with this definition, $\bar{S}(\boldsymbol{k})$ is left invariant under a particle-hole transformation of the partially-filled Landau level ($\bar{\rho}(\boldsymbol{k}) \to -\bar{\rho}(-\boldsymbol{k})$ for $k \neq 0$ under particle-hole transformation). With this definition, contributions from more than one partially-occupied Landau level are additive, and in our opinion, its use greatly clarifies the treatment of such systems.

In a system with Landau quantization of the electronic states into macroscopically-degenerate Landau levels, where levels with index $n < n_0$ are filled ($v_n = 1$), levels with index $n > n_0$ are empty ($v_n = 0$), and the level with index $n_0$ is partially-filled ($0 < v_{n_0} < 1$), the conventional structure factor (1) and the guiding center structure factor (6) are related by

$$(\textstyle\sum_n v_n)(S(\boldsymbol{k}) - 1) = |f_{n_0,n_0}(\boldsymbol{k})|^2 (\bar{S}(\boldsymbol{k}) - \bar{S}(\infty)) - \sum_{n,n'} v_n v_{n'} |f_{n,n'}(\boldsymbol{k})|^2 . \tag{9}$$

Here $f_{nn'}(\boldsymbol{k})$ depends on the structure of the orbits of the Landau levels:

$$f_{nn'}(\boldsymbol{k}) = \langle n, 0 | e^{-i\boldsymbol{k}\cdot\tilde{\boldsymbol{R}}} | n', 0 \rangle , \tag{10}$$

where $|n, 0\rangle$ is a guiding-center coherent state in the Landau level with index $n$, and generally falls off rapidly in an exponential (Gaussian) fashion at large $|\boldsymbol{k}|$, $\lim_{\lambda\to\infty} f_{nn'}(\lambda\boldsymbol{k}) \to 0$. This form of the structure function only requires the assumption that translation and (2D) inversion symmetry (180° rotation in the 2D plane) are unbroken. It is valid in a FQH ground state, and remains valid at finite temperatures where $k_B T$ is much smaller that the energy gaps between Landau levels. Here $\lim_{\lambda\to\infty} \bar{S}(\lambda\boldsymbol{k}) \to \bar{S}(\infty) = v_0(1 - v_0)$, assuming particles in the partially-filled level are fermions (if they were bosons, it would be $v_0(1 + v_0)$). If the energy scale of interactions between particles in the partially-occupied level is much smaller than energy gaps between Landau levels, $\bar{S}(\boldsymbol{k}) - \bar{S}(\infty) \to 0$ when the temperature is high enough so that guiding-center degrees of freedon become completely disordered without affecting the Landau-orbit degrees of freedom. Also $\bar{S}(\boldsymbol{k}) \to 0$ in the limit that $v_0$ becomes 0 (empty) or 1 (filled), and is only a property of partially-filled levels. A key property, discovered by GMP, is that in the gapped incompressible FQH ground states

$$\lim_{\lambda\to 0} \bar{S}(\lambda\boldsymbol{k}) \propto \lambda^4 , \tag{11}$$

which is strictly a ground-state property.

At finite temperatures $T$ in the grand canonical ensemble, $\bar{S}(0)$ is finite if there are no long-range interactions between the guiding-centers of the particles, with a value related to the variance of charge fluctuations in the equilibrium state; if there are long-range interactions where the Fourier transform of the interaction potential diverges at small $\boldsymbol{k}$ as $\tilde{V}(\lambda\boldsymbol{k}) \propto \lambda^{-\alpha}$ with $0 < \alpha \leq 2$ as $\lambda \to 0$, a neutralizing background and the canonical ensemble are required, and use of the random-phase approximation (RPA) at long wavelengths shows that $\bar{S}(\lambda\boldsymbol{k})$ vanishes as $\lambda^\alpha$ in this limit when $T > 0$. At $T = 0$, it can be shown that $\bar{S}(\lambda\boldsymbol{k})$ must vanish as $\lambda^\gamma$ with $\gamma > 2$ if 2D inversion and translation symmetry are unbroken, and if the system is incompressible (gapped), $\bar{S}(\boldsymbol{k})$ must be analytic and even in the components of $\boldsymbol{k}$, consistent with the GMP result $\gamma = 4$ for FQH states, where

$$\bar{S}(\lambda\boldsymbol{k}) = \lambda^4 \bar{\Gamma}^{abcd} k_a k_b k_c k_d \ell^4 + O(\lambda^6) . \tag{12}$$

Here $\bar{\Gamma}^{abcd}$ is a rank-4 tensor with the symmetries $\bar{\Gamma}^{abcd} = \bar{\Gamma}^{cdab} = \bar{\Gamma}^{bacd}$, although only its fully-symmetrized form is detected by the small-$\boldsymbol{k}$ behavior of $\bar{S}(\boldsymbol{k})$. Further, $\ell \equiv \sqrt{\hbar/|eB|}$ is the magnetic length.

While unbroken 2D inversion symmetry (discrete two-fold rotation symmetry) is the only fundamental point symmetry of the uniform FQH state (which resists electric polarization with

an energy gap for excitations carrying electric dipole moment), most authors study the FQH in model systems where 2D inversion symmetry is promoted to a continuous $SO(2)$ rotation symmetry where the generator (azimuthal angular momentum) is

$$L = \tfrac{1}{2}(eB)\sum_i g_{ab}(\bar{R}_i^a \bar{R}_i^b - \tilde{R}_i^a \tilde{R}_i^b),  \tag{13}$$

where $g_{ab}$ is a Euclidean-signature metric tensor (symmetric positive definite, with $\det g = 1$). While this can be justified as an "emergent" low-energy symmetry in systems where the underlying atomic-scale crystal structure has square or hexagonal symmetry in the 2D plane, there is no fundamental place for such a symmetry in the physics of solid (crystalline) condensed matter systems, so it should be regarded as a "toy model" feature that simplifies treatment of the system. For FQH systems with such a symmetry, $\bar{S}(\mathbf{k})$ must be an analytic function of $k^2 = g^{ab}k_a k_b$, where $g^{ab}$ is the inverse metric, and in the small-$k^2$ limit,

$$\bar{S}(\mathbf{k}) = \bar{S}_4(k^2\ell^2)^2 + \bar{S}_6(k^2\ell^2)^3 + O((k^2\ell^2)^4).  \tag{14}$$

The rest of this paper will be concerned with the coefficients $\bar{S}_4$ and $\bar{S}_6$ of this expansion in such "toy model" FQH states with $SO(2)$ continuous rotational symmetry in the 2D plane.

## 3  Composite-boson picture of FQHE

In this section, we review the composite-boson formulation of FQH states [24, 27–31]. As we'll see, one obtains a natural separation between the guiding center and Landau orbit degrees of freedom that allows one to differentiate the physics within a Landau level from the inter-Landau level one. The topological properties associated with composite-bosons are used to re-express the quantities that appear in Haldane bound, making Landau level projection and particle-hole transformation transparent. A reader interested in the results of numerical computations may skip to the next section.

The "composite boson", made of $p$ particles "bound to $q$ flux quanta", is an elementary using of FQH states, which means that they occupy (essentially exclusively) a region of the 2D plane through which $q$ London flux quanta pass, and which supports $q$ independent one-particle states within the Landau level [24, 27–31]. The partial filling factor $\nu_{n_0}$ is $p/q$, and the charge on the elementary excitation is $e/q = (pe)/k$ where $pe$ is the charge of the elementary unit, and $k = pq$ is the level index of an emergent $U(1)_k$ Abelian Chern-Simons gauge field that describes the Berry phases that are generated when the size-$q$ "hole" carrying the $p$ electrons moves through the incompressible FQH background. The exchange of two such "holes" produces a Berry phase $(-1)^k$, which must cancel the exchange phase $(-1)^p$ of the two groups of $p$ fermions that respectively populate the two holes, so that the "flux-charge composites" behave like bosons when exchanged, leading to the selection rule $(-1)^k = (-1)^p$ (or $(-1)^k = 1$ if the particles of the fluid were bosons, as in some "toy-model" states). The fluid in the partially-filled Landau level contributes an amount $(pe)^2/2\pi\hbar k$ to the Hall conductance of the system; completely-filled Landau levels can also be consistently described in this "bosonic" way with $p = k = 1$.

In addition to the two basic topological numbers $p$ and $q$, an independent topological number is the *chiral central charge* $c_-$ (a signed quantity) which is a quantum anomaly of the (Virasoro) algebra obeyed by the 1D momentum density $\pi_a(x)$ (or equivalently, using the relation $\pi_a(x) = F_{ab}P^b(x)$, the 1D electric polarization $P^a(x)$) of the gapless edge degrees of freedom of the FQH fluid: if $\pi(x) = t^a(x)\pi_a(x)$, where $\mathbf{t}(x)$ is the tangent unit vector of the edge, this algebra is

$$[\pi(x), \pi(x')] = i\hbar\delta'(x-x')\pi(x) + \tfrac{1}{12}i\hbar^2 c_- \delta'''(x-x'),  \tag{15}$$

where $\delta(x)$ is a 1D Dirac delta function on the edge parameterized by $x$, and $\delta'(x)$ is its first derivative, *etc*. The anomaly $c_-$ can be written

$$c_- = \sum_n \nu_n + \bar{c}\,, \tag{16}$$

where $\bar{c}$ is an extra contribution from the partially-filled Landau level, which is odd under a particle-hole transformation, and vanishes when the level is filled or empty; a filled Landau level contributes $\nu_n = 1$ to the total $c$, and an empty level does not contribute. As an example, for the $(p,q) = (1,3)$ $\nu = \frac{1}{3}$ Laughlin state in the lowest Landau level, $\bar{c} = \frac{2}{3}$, $c_- = \frac{1}{3} + \frac{2}{3} = 1$, while for the $(p,q) = (2,3)$ $\nu = \frac{2}{3}$ "anti-Laughlin state" (particle-hole-transformed Laughlin state), $\bar{c} = -\frac{2}{3}$, $c_- = \frac{2}{3} - \frac{2}{3} = 0$.

In the special case of $SO(2)$ continuous rotational symmetry, a geometrical property of the FQH fluid becomes "topological" when azimuthal angular momentum becomes quantized. An electron in a standard Landau level labeled by $n = 0, 1, 2 \ldots$ has Landau-orbit angular momentum $\hbar \tilde{s}_n$ relative to its guiding center, given by the second term (involving the Landau-orbit vectors i.e. $\tilde{L} = \sum_i g_{ab} \tilde{R}_i^a \tilde{R}_i^b$) in (13), with $\tilde{s}_n = -(n + \frac{1}{2})$, and the "guiding-center spin" $S_{GC}$ of the elementary unit (composite boson) is found by computing the guiding center contribution to its angular momentum and subtracting the value $\frac{1}{2}\hbar k = \frac{1}{2}\hbar pq$ which it would have had if each of the $q$ orbitals had uniform filling $\nu_{n_0} = p/q$. With this definition, $S_{GC}$ is odd under particle-hole transformations of the partially-filled Landau level, and vanishes in the limit that the level is empty or filled, and is also "correctly" quantized so $2S_{GC}$ is an integer, with $(-1)^{2S_{GC}} = (-1)^p (-1)^k$. Because of the composite-boson selection rule that $(-1)^k = (-1)^p$ for fermionic systems, $S_{GC}$ is always an integer when the charge-$e$ particles are fermions, but can be half-odd-integral in models where they are bosons, as in the $\nu = \frac{1}{2}$ Laughlin state, where $S_{GC} = -\frac{1}{2}$. Note that when the *total* spin of the composite boson is obtained by adding its Landau orbit spin to the guiding-center spin, one obtains

$$(-1)^{2S_{GC}}(-1)^{2p\tilde{s}_{n_0}} = (-1)^{2S_{GC}}(-1)^p = (-1)^k\,, \tag{17}$$

so the composite-boson selection rule ensures that the total (azimuthal) spin of the composite boson is half-odd-integral in fermionic systems, and integral in bosonic systems.

The usual way to express these quantities (which in our view fails to properly distinguish Landau orbit from guiding center quantities) is by the so-called "shift" which has been defined [32] using spherical geometry as

$$\mathcal{S} = \nu^{-1}N - N_\Phi\,, \tag{18}$$

where the uniform FQH state is placed on a sphere enclosing a Dirac magnetic monopole emitting an integer $N_\Phi$ London flux quanta. In the spherical geometry, there is continuous azimuthal rotational symmetry about any axis normal to the surface of the sphere. On a flat compact surface (the 2D torus) the number of independent orbitals in each Landau level is $N_\Phi$; for standard non-relativistic Landau levels on the uniformly curved sphere this is modified to $N_\Phi + 2|\tilde{s}_n|$ where $\hbar \tilde{s}_n$ is given by the Landau orbit part of (13): $\tilde{s}_n = -(n + \frac{1}{2})$, $n = 0, 1, 2, \ldots$. For the integer QH fluid, where $\nu_n = 0$ or $1$,

$$-\tfrac{1}{2}\nu\mathcal{S} = -\sum_n \nu_n |\tilde{s}_n| = \sum_n \nu_n \tilde{s}_n\,, \tag{19}$$

consistent with Read's interpretation [5] of $-\frac{1}{2}\mathcal{S}$ as the "mean orbital spin per particle" (multiplying this by $\nu$ makes it the "mean orbital spin per flux quantum").

The composite boson with guiding-center angular momentum $S_{GC}$ occupies $q$ orbitals, so its contribution to the spin per flux quantum (*i.e*, per orbital) is $q^{-1}S_{GC}$, and the relation of

$S_{\text{GC}}$ to the "shift" is

$$S_{\text{GC}} = -q \sum_n \nu_n (\tfrac{1}{2}\mathcal{S} + \tilde{s}_n) = q \sum_n \nu_n (n - \tfrac{1}{2}(\mathcal{S} - 1)). \tag{20}$$

As examples, consider the Jain FQH states with $q = pr + 1$ where $p, r \in \{1, 2, \cdots\}$ and $\nu = p/q$. These correspond to an elementary composite-boson unit with the following filling sequence of the LL orbitals: $(10^{r-2})^p 0^{p+1}$ for $r \geq 2$ and $p0^p$ for $r = 1$.[1] Here the numbers represent the number of particles in the orbital, and the superscript '$j$' refers to $j$-fold repetition. Moreover, $m^{\text{th}}$ orbital contributes $m + 1/2$ times its net charge (number of electrons in the orbital minus $p/q$) to the guiding center spin which for the Jain sequence is $2S_{\text{GC}} = -p(p+r-1)$. The "anti-Jain" FQH states at $\nu = 1 - \frac{p}{pr+1}$ have $2S_{\text{GC}} = p(p+r-1)$. For fermionic states, these values apply not just to a Laughlin state in the lowest Landau level, but also to a Laughlin state in *any* partially occupied level, where all lower-energy Landau levels are fully occupied. This illustrates why we consider that the guiding-center spin, rather than the "shift", is the appropriate quantity for characterizing rotationally-invariant FQH states.

A long-standing issue in the quantum Hall effect is the question of its relation to two-dimensional conformal field theory. In the years after its discovery, it became clear that its topologically-protected gapless edge degrees of freedom (or, more generally, the gapless edge (1+1)-dimensional degrees of freedom at the boundary between two regions supporting topologically-distinct incompressible quantum Hall states) are defined in a low-energy Hilbert space that contains a unitary representation of the Virasoro algebra (15). However the low energy Hamiltonian of the edge states does *not* exhibit conformal invariance as edge excitations do not in general propagate with a universal Lorentz speed. The low-energy structure of the gapped bulk FQH state is described by a topological quantum field theory (TQFT) which describes the braiding properties of its topological excitations. There is a compatibility condition between the bulk TQFT and the edge Virasoro algebra in that the TQFT Braiding relations fix $c_-$ mod 8, but do not fix $c_-$ itself [33].

# 4 Haldane bound for isotropic FQH states and sixth order term in the static structure factor

In the following sections, we investigate the conditions under which the Haldane bound is saturated. In appendix B, we show that $SO(2)$ rotational invariance is required to make the fluctuations of angular momentum vanish. Moreover, the isotropic Jain states at $\nu = p/(2np-1)$ have been found to not saturate the Haldane bound in previous studies [11–13]. The primary reason for this is that these states contain more than one graviton mode that have both spin $+2$ and $-2$ chiralities in the long wavelength limit. Moreover, these states do not satisfy the weak maximal chirality conditions, i.e., low lying part of their entanglement spectra are not fully chiral owing to the presence of counter-propagating edge modes. Starting with this section, we specialize to at least weakly maximally chiral and isotropic FQH ground states, since they appear to be the largest set of possible candidates that could saturate the Haldane bound.

The Haldane bound can be stated in terms of the guiding center contribution to Hall viscosity $\bar{\eta}_H$, in the presence of $SO(2)$ rotational invariance, as follows [2]:

$$\bar{S}_4 \geq \pi \left| \frac{\bar{\eta}_H}{eB} \right|. \tag{21}$$

---

[1] These root configurations are the uniform versions of the ones in Ref. [50, 51].

$$\bar{\eta}_H = -\frac{eB}{4\pi}\frac{S_{\mathrm{GC}}}{q} \tag{22}$$

$$= \frac{eB}{8\pi}\nu(\mathcal{S}-1)\,, \tag{23}$$

where we have assumed that only the lowest Landau level is filled (partially), and $\nu = p/q$. The guiding center contribution to Hall viscosity $\bar{\eta}_H$ is expressed in terms of the guiding center spin of the composite-boson $S_{GC}$ as defined in the section 3 and comes purely from correlations within the lowest Landau level. Since $S_{GC}$ is odd under particle-hole (PH) symmetry, the expression of Haldane bound is PH symmetric. In the last line, we have expressed it using the conventional Wen-Zee shift $\mathcal{S} = \nu^{-1}N - N_\phi$ [32].

The physics underlining the bound is that $\bar{S}_4$ measures the total spectral weight of spin-2 or quadrupolar excitations generated by the guiding center density operator [1,25]. These excitations can be regarded as the long-wavelength fluctuations of area-preserving strains [26]. $\bar{S}_4$ corresponds to the quantum metric in the parameter space of area-preserving linear deformations, while $\bar{\eta}_H$ corresponds to the Berry curvature. The general inequality between the two then yields the bound. We refer the reader to appendix A for an explicit derivation.

A second property of $\bar{S}(\boldsymbol{k})$ has been conjectured in the literature using field theoretic approaches. [16–20] It states that $\bar{S}_6$, for chiral FQH states, contains information about the chiral central charge of edge theory and is given by:

$$\bar{S}_6 = \pm\frac{\nu\left(3\bar{\mathcal{S}}^2+1\right)-\bar{c}+12\nu\,\mathrm{var}(s)}{96} \tag{24}$$

$$= \pm\frac{12\nu\left((S_{\mathrm{GC}}/p)^2+\mathrm{var}(s)\right)-\bar{c}}{96}\,, \tag{25}$$

where $\bar{c} \equiv c_- - \nu$ is odd under PH transformation and $c_-$ is the chiral central charge of the edge theory defined as the central charge of right-movers minus that of left-movers. It is the quantum anomaly that appears in the Virasoro algebra obeyed by edge state momentum density $\pi_a(x)$ as in Eq. (15). Moreover, $\mathrm{var}(s) \equiv \frac{s^T K^{-1} s}{t^T K^{-1} t} - \left(\frac{s^T K^{-1} t}{t^T K^{-1} t}\right)^2$ is an additional quantity conjectured to occur in the case of multi-component Abelian FQH states which has been called the "orbital spin variance" [34,35]. Here $K$ corresponds to the $K$-matrix of the FQH state with $t$ and $s$ representing the charge and spin vectors respectively [36]. The combination $\nu\left((S_{\mathrm{GC}}/p)^2+\mathrm{var}(s)\right) = (s-t/2)^T K^{-1}(s-t/2)$ is odd under PH transformation. At present, we are not aware of its interpretation in the composite-boson framework. The $\pm$ sign is included to make $\bar{S}_6$ even under PH transformation. Chiral FQH states and their PH conjugates correspond to $+$ and $-$ signs respectively.

We compare this conjecture with our results for the Moore-Read and $\nu = 2/5$ Jain-like FQH ground state where we are not aware of any previous numerical calculation for $\bar{S}_6$ (although see [23] for an analytical calculation for Moore-Read wavefunction). Importantly, we will show that for weak maximally chiral FQH states, $\bar{S}_6$ is bounded above by the theoretical prediction.

# 5 Methods and numerical results

We consider spin-polarized FQH states in the lowest Landau level. Let us assume that the energy gap to excitations to the first Landau level is large compared to the energy scale of interactions and filling fraction $\nu \leq 1$ so that we can project to the lowest Landau level.

We utilize the infinite cylinder geometry with its axis aligned along the $x$-direction and circumference of length $L_y$ in the periodic $y$-direction. To this end, we use the commutation relation $[\bar{R}^a, \bar{R}^b] = -i\frac{\hbar}{eB}\epsilon^{ab}$ to define the single-particle guiding center momentum operators

$\bar{P}_a = -eB\epsilon_{ab}\bar{R}^b$. The eigenstates of $\bar{P}_y$ operator can be used to construct the single-particle Hilbert space of the LLL spanned by $|n\rangle$. Assuming periodic boundary conditions along $y$-direction, i.e. $\mathcal{T}_y(L_y)|n\rangle = e^{i\frac{\bar{P}_y L_y}{\hbar}}|n\rangle = |n\rangle$, we obtain $\bar{P}_y|n\rangle = \hbar n\kappa|n\rangle$ with $\kappa = 2\pi/L_y$ and $n \in \mathbb{Z}$.

To obtain a second-quantized form for the operators, we define the fermion operators $c_n, c_n^\dagger$ that act on the orbital labeled by '$n$'. The guiding center density operator becomes:

$$\bar{\rho}(\boldsymbol{k}) = \delta_{m\kappa,k_y} e^{-\frac{ik_x k_y \ell^2}{2}} \sum_n e^{-in\kappa k_x \ell^2} c_n^\dagger c_{n+m}. \tag{26}$$

The long-wavelength expansion of the guiding center static structure factor is obtained by setting $k_y = 0$ and expanding in powers of $k_x \ell$ so that

$$\bar{S}(k_x) = \sum_{n=2}^{\infty} \bar{S}_{2n} \times (k_x \ell)^{2n}, \tag{27}$$

$$\bar{S}_{2n} = \frac{(-1)^n}{N_\Phi(2n)!} \sum_{m,l} (m\kappa\ell)^{2l} \langle\Psi|\delta\hat{n}_l \delta\hat{n}_{l+m}|\Psi\rangle, \tag{28}$$

where $\delta\hat{n}_m \equiv \hat{n}_m - \langle\Psi|\hat{n}_m|\Psi\rangle$ and $\hat{n}_m \equiv c_m^\dagger c_m$. This subtraction is crucial for numerical stability. In practice, the sum over '$l$' can be restricted to the unit cell of the translationally invariant matrix-product state. Moreover, we employ a cutoff in the sum over '$m$' so that $|m| \leq m_{\max}$. We choose $m_{\max}\kappa\ell^2 \gg \xi_x$, where $\xi_x$ is the correlation length of the ground state in the $x$-direction, such that $\bar{S}_{2n}$ become independent of the cutoff. Note that if in addition $L_y \gg \xi$, $\bar{S}_{2n}$ become independent of $L_y$ and converge to their values in the 2D limit.

The Hamiltonian of the electrons in the lowest Landau level can be expressed using the guiding center density operators. In this paper, we use density-density interactions that have the form:

$$H = \frac{1}{2L_y} \sum_{k_y} \int \frac{dk_x}{2\pi} V(\boldsymbol{k}) e^{-g^{ab}k_a k_b \ell^2/2} : \bar{\rho}(\boldsymbol{k})\bar{\rho}(-\boldsymbol{k}) :, \tag{29}$$

where $V(\boldsymbol{k})$ is the two-particle interaction potential with $V(\boldsymbol{k}) = 2\pi/k$ for Coulomb interactions and $V(\boldsymbol{k}) = 2L_n((k\ell)^2)$ for the $n^{\text{th}}$ Haldane pseudopotential, and $g^{ab} = \delta^{ab}$ is the inverse Euclidean metric tensor. We modify the Coulomb interaction to have a Gaussian envelope, i.e. $V(r) = \frac{1}{r}e^{-r^2/2\xi^2}$ with $\xi = 6\ell$. This value was previously found to not affect results significantly [37].

We obtain the FQH ground states at $\nu = 1/3, 1/5$, and $2/5$ using iDMRG and compute $\bar{S}_{2n}$ via Eq. (28). Additionally, we use the matrix-product state construction of the Moore-Read state obtained via conformal field theory in Ref. [21]. Large sizes ($L_y \gg \xi$) have been achieved so that $\bar{S}_4$ and $\bar{S}_6$ converge to their values in the 2D-limit. The plots of $\bar{S}_4$ and $\bar{S}_6$ versus inverse circumference, i.e. $\ell/L_y$, are presented in Fig. 1. Below, we provide an interpretation of these results in the context of the lower bound of Eq. (21), and the theoretical prediction of Eq. (25).

## 5.1  $\nu = 1/3$

Fig. 1 (a) and (b) show $\bar{S}_4$ and $\bar{S}_6$ at $\nu = 1/3$ for $V_1$-pseudopotential and Coulomb interaction respectively. In both cases, the two quantities become independent of cylinder circumference at large sizes and accurately capture the physics in 2D-limit. In the former case, the Laughlin wavefunction [6] is an exact zero energy ground state [38] and $\bar{S}_4$ saturates the Haldane bound. Moreover, $\bar{S}_6$ agrees with Eq. (25) at large sizes. These facts are well known and confirm the accuracy of our numerical procedure [1,39,40]. On the other hand, for Coulomb

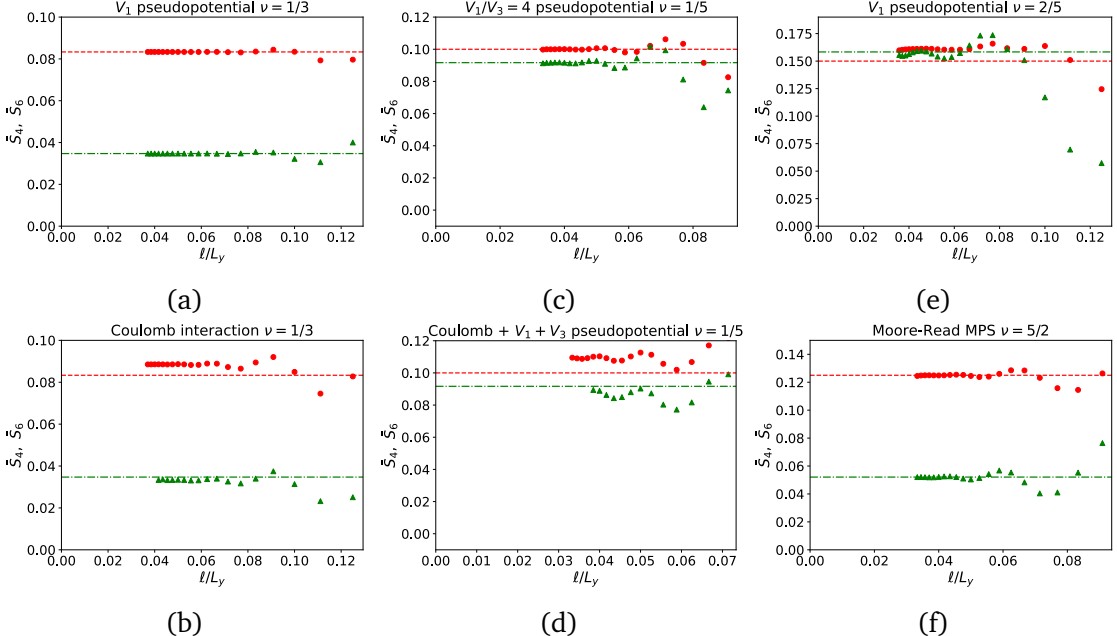

Figure 1: $\bar{S}_4$ (red circles) and $\bar{S}_6$ (green triangles) vs. inverse circumference for various FQH states where $\bar{S}(k_x) = \bar{S}_4(k_x\ell)^4 + \bar{S}_6(k_x\ell)^6 + \cdots$. The red-dashed line corresponds to the Haldane bound for $\bar{S}_4$ (see Eq. (21)) and green dash-dotted line corresponds to the theoretical prediction for $\bar{S}_6$ (see Eq. (25)). The maximum MPS bond-dimension in (a) and (c) is $\chi = 8192$, and $\chi = 16384$ in (b), (d) and (e). In (f), the maximum number of CFT levels is $N_{\text{CFT}} = 20$. The presented results are nearly independent of the bond-dimension. In (d), we have added pseudopotentials $V_1 = 1$, $V_3 = 0.5$ to Coulomb interaction.

interactions, $\bar{S}_4$ is approximately 6% bigger than the bound demonstrating that Haldane bound is not saturated. Moreover, $\bar{S}_6$ is around 4% below the value predicted by theory.

This relatively large disparity between the Laughlin wavefunction and Coulomb ground state should be contrasted with the near 99% overlap between the two in spherical geometry upto 15 electrons (for example see [41]). An important difference between the two measures is that the wavefunction overlaps go to zero exponentially in size while $\bar{S}_4, \bar{S}_6$ obtained here are size independent. Moreover, the overlap is a net measure of all possible differences between two wavefunctions while the long-wavelength correlation function is just one physical property. For these reasons, we believe that the two can play complementary roles when comparing different wavefunctions.

## 5.2 $\nu = 1/5$

The results in Fig. 1 (c) and (d) obtained for the $\nu = 1/5$ FQH state present a picture similar to $\nu = 1/3$. In (a), we have used $V_1$-$V_3$ Haldane pseudopotentials with $V_1 = 4V_3$ and our findings agree with the previously known results. In contrast, we find in 1 (d) that $\bar{S}_4$ for the Coulomb plus $V_1$-$V_3$ interaction is approximately 9% above the bound. Moreover, $\bar{S}_6$ is approximately 2.5% below the prediction at the largest indicated circumference, though the latter may contain bigger finite size effects.

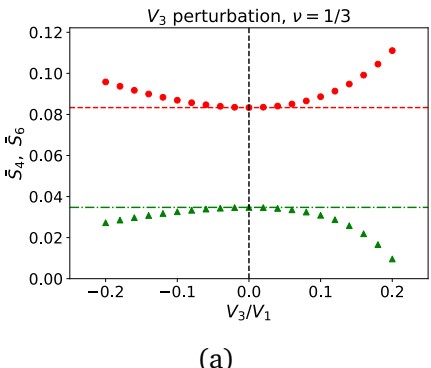
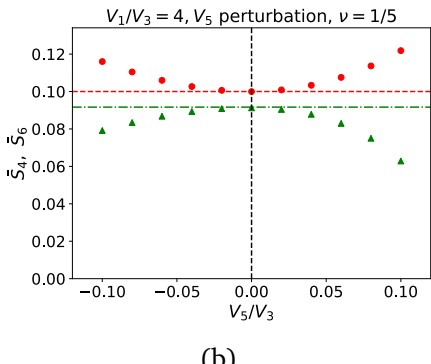

(a)                                                (b)

Figure 2: $\bar{S}_4, \bar{S}_6$ at (a) $\nu = 1/3$ and (b) $\nu = 1/5$ for (a) $V_3$ perturbation around the $V_1$ pseudopotential (b) $V_5$ perturbation around $V_1 - V_3$ pseudopotential interaction. The cylinder circumference is fixed at $L_y = 24\ell$.

Table 1: $\Delta\bar{S}_4$: deviation of $\bar{S}_4$ from the lower bound of Eq. 21. $\Delta\bar{S}_6$: deviation of $\bar{S}_6$ from the field theory prediction of Eq. 25. The values correspond to the numerical results of Fig. 1 at the largest sizes.

| FQH state | Interaction | $\Delta\bar{S}_4$ | $\Delta\bar{S}_6$ |
|---|---|---|---|
| Laughlin-1/3 | $V_1$ | $\approx 0\%$ | $\approx 0\%$ |
| | Coulomb | $\approx 6\%$ | $\approx -4\%$ |
| Laughlin-1/5 | $V_1/V_3 = 4$ | $\approx 0\%$ | $\approx 0\%$ |
| | Coulomb+$V_1$+$V_2$ | $\approx 9\%$ | $\approx -2.5\%$ |
| Jain-2/5 | $V_1$ | $\approx 7\%$ | $\approx -1\text{-}2\%$ |
| Moore-Read | MPS (3-body) | $\approx 0\%$ | $\approx 0\%$ |

## 5.3   $\nu = 2/5$

In (e), we plot the data for the ground state at $\nu = 2/5$ in the presence of $V_1$-Haldane pseudopotential interaction. This state is usually interpreted as an integer quantum Hall state of composite-fermions (CFs) at filling fraction $\nu_{\text{cf}} = 2$ [14], where the CFs are constructed by attaching two flux quanta to electrons. Ref. [42] provided a confirmation of the Jain FQH-like ground state in the infinite cylinder geometry through the computation of its Hall viscosity, chiral central charge, and topological spin. Although the ground state of $V_1$ pseudopotential is not identical to Jain wavefunction at $\nu = 2/5$, it has the same topological properties.

$\nu = 2/5$ is the only filling fraction we analyze where the orbital spin variance term in Eq. (25) is non-zero. At large sizes, numerically computed $\bar{S}_6$ is extremely close but about 1-2% below the theoretical prediction. However, we remark that the $\nu = 2/5$ ground state for $V_1$-pseudopotential interaction cannot be expressed as a conformal block wavefunction. This is apparent in the fact that $\bar{S}_4$ is around 7% above the Haldane bound at the largest sizes. Moreover, in (b) and (d) where we did not use the model Hamiltonians, $\bar{S}_6$ was found to be close but not equal to the conjectured expression. As such, the evidence for Eq. (25) must be interpreted with caution. Nevertheless, we propose below that Eq. (25) could be valid when the relation is modified to an upper bound on $\bar{S}_6$. A comparison of our results with CF model wavefunctions could provide useful insights in this matter [43].

## 5.4 Moore-Read state

In (f), we use the matrix-product state construction of the Moore-Read state [21,22] to obtain the long-wavelength expansion of its static structure factor via Eq. (28). As is clear from the figure, it saturates Haldane bound [10] and $\bar{S}_6$ is in excellent agreement with the theoretical prediction at large sizes. This suggests that Eq. (25) is valid for all model wavefunctions obtained via the conformal blocks approach [7]. We comment that similar result was obtained in Ref. [23] using a more direct conformal-field theory approach by generalizing the Moore-Read state to curved backgrounds.

The deviations of $\bar{S}_4$ and $\bar{S}_6$ from the lower bound in Eq. (21) and the theoretical prediction in Eq. (25) respectively are summarized in Table 1.

## 6 Perturbations around model Hamiltonians and behavior of $\bar{S}_6$

An interesting observation can be made from the results of Fig. 1 (b), (d) and (e). $\bar{S}_6$ is close but below the theoretical prediction for FQH ground states of Hamiltonians that deviate away from the model Hamiltonians. This suggests that $\bar{S}_6$ is bounded above by the value in Eq. (25). For a more transparent analysis, we perturb the Laughlin FQH ground states at $\nu = 1/3$ and $\nu = 1/5$ by introducing $V_3$ and $V_5$ pseudopotentials respectively on top of the model Hamiltonians. The results are presented in Fig. 2. It can be seen that $\bar{S}_6$ has its maximum value at the point of zero perturbation, suggesting an upper bound. Also, note that $\bar{S}_4$ increases above the minimum value allowed by the Haldane bound upon applying the perturbation, confirming the observations of Fig. 1.

## 7 Discussion

In summary, we have computed the coefficients of $(k\ell)^4$ and $(k\ell)^6$ terms in the long-wavelength expansion of guiding center static structure factor. We observed that the Haldane bound is not saturated for fractional quantum Hall (FQH) ground states upon deviating away from the model wavefunctions, even at relatively simple filling fractions such as $\nu = 1/3, 1/5$. This suggests that the minimal field theory descriptions must be supplemented with additional terms or degrees of freedom to describe general FQH states.

As shown in appendix B, the saturation of Haldane bound requires rotational invariance. Moreover, based on the arguments in appendix and the spectral sum rules derived in Ref. [25], the spin-2 graviton excitation must be maximally chiral [11–13,44,45]. Our calculations imply that except for ideal conformal block wavefunctions, no other FQH state may be expected to have perfectly chiral graviton mode in the thermodynamic limit.

From a practical point of view, the chirality of graviton mode in experiments, such as circularly polarized Raman scattering [25,44,46–48], has been proposed to be a possible way of distinguishing different topological orders. We have been able to reach large sizes where the finite size corrections are essentially absent. Thus, our calculations can be used to predict the degree of graviton chirality in the 2D limit. To demonstrate this, we consider the value of $\bar{S}_4$ for the isotropic FQH ground state of Coulomb interactions at $\nu = 1/3$. It is found to be about 6% above the Haldane bound. Using the spectral sum rules of Ref. [25] and appendix C, we predict the ratio of intensities of the two graviton chiralities, defined as in Ref. [25], to be $\approx 0.03$. However, this may be modified by effects such as quantum well width, Landau level mixing, and the presence of impurities.

The conformal block wavefunctions, that saturate the bound, satisfy maximal chirality. Hence both their graviton modes and entanglement spectra contain only one chirality. Simultaneously, the generic FQH states, which do not saturate the Haldane bound, have both chiralities in the graviton modes as well as the entanglement spectra at least above the entanglement gap. Since the former corresponds to bulk and latter to edge properties, this appears to be a feature of the bulk-boundary correspondence. The precise relations between the two are not understood at present. Further numerical exploration would require finite-size scaling studies of the time-dependent correlation functions of an operator that is sensitive to the graviton chirality [12, 45, 49].

We have additionally found that the effective field theory conjecture [16–20] for the value of $\bar{S}_6$, the coefficient of $(k\ell)^6$ in the long-wavelength expansion of the guiding center static structure factor, does not hold as an equality for generic FQH states. Nevertheless, our numerical experiments indicate that it could act as an upper bound for $\bar{S}_6$. A theoretical justification of this conjecture is an interesting problem for future work.

# Acknowledgments

We thank R. N. Bhatt and Matteo Ippoliti for discussions. The iDMRG and Moore-Read MPS numerical computations were carried out using libraries developed by Roger Mong, Michael Zaletel and the TenPy collaboration.

**Funding information** This research was partially supported by NSF through the Princeton University (PCCM) Materials Research Science and Engineering Center DMR-2011750. Additional support was received from DOE BES Grant No. DE-SC0002140.

# A Derivation of the Haldane bound

In this appendix, we provide a pedagogical derivation of the Haldane bound. [2] As mentioned in the main text, it arises from a general bound between the quantum metric and Berry curvature. It will be shown that the former relates to $\bar{S}_4$, while the latter corresponds to Hall viscosity.

We construct the regularized guiding center density operator in the thermodynamic limit: $\delta\bar{\rho}(\boldsymbol{k}) = \bar{\rho}(\boldsymbol{k}) - 2\pi\nu\ell^{-2}\delta^{(2)}(\boldsymbol{k})$. It satisfies the GMP algebra since $\bar{\rho}(\boldsymbol{k} = 0)$ does not enter the commutation relation [1, 2]:

$$[\delta\bar{\rho}(\boldsymbol{k}), \delta\bar{\rho}(\boldsymbol{p})] = 2i\sin\left(\frac{\epsilon^{ab}k_a p_b \ell^2}{2}\right)\delta\bar{\rho}(\boldsymbol{k} + \boldsymbol{p}), \tag{A.1}$$

where we have assumed a uni-modular metric, i.e. $\det(g) = 1$. Let's expand $\delta\bar{\rho}(\boldsymbol{k})$ as follows:

$$\delta\bar{\rho}(\boldsymbol{k}) = \sum_{n=1}^{\infty}(-i\ell)^n \bar{\rho}_n^{a_1 a_2 \cdots a_n} k_{a_1} k_{a_2} \cdots k_{a_n}, \tag{A.2}$$

where $\bar{\rho}_n^{a_1 a_2 \cdots a_n}$ are hermitian and fully symmetric in the upper indices. We define $\delta\bar{P}_a \equiv -\hbar\ell^{-1}\epsilon_{ab}\rho_1^b$ to be the regularized generators of translations since $e^{ix_0^a \delta\bar{P}_a/\hbar}\delta\bar{\rho}(\boldsymbol{k}) = e^{-ix_0^a k_a}\delta\bar{\rho}(\boldsymbol{k})e^{ix_0^a \delta\bar{P}_a/\hbar}$. Moreover, they commute, i.e. $[\delta\bar{P}_a, \delta\bar{P}_b] = 0$ and can be chosen to annihilate a translationally invariant ground state, i.e., $\delta\bar{P}_a|0\rangle = 0$.

The static structure factor of Eq. (6) can be written as follows:

$$\bar{S}(\boldsymbol{k}) = \frac{1}{N_{\text{orb}}} \langle \delta\bar{\rho}(\boldsymbol{k}) \, \delta\bar{\rho}(-\boldsymbol{k}) \rangle_0 \,, \tag{A.3}$$

where the expectation value is taken with respect to the ground state $|0\rangle$ and $N_{\text{orb}}$ is the number of orbitals on the spherical geometry. Its series expansion is given by the following expression:

$$\bar{S}(\boldsymbol{k}) = \frac{1}{N_{\text{orb}}} \sum_{nl} (i\ell)^n (-1)^l k_{a_1} \cdots k_{a_n} \left\langle \delta\bar{\rho}_{n-l}^{a_1\cdots a_{n-l}} \delta\bar{\rho}_l^{a_{n-l+1}\cdots a_n} \right\rangle_0 \,, \tag{A.4}$$

where $\delta\bar{\rho}_n^{a_1 a_2 \cdots a_n} \equiv \bar{\rho}_n^{a_1 a_2 \cdots a_n} - \langle \bar{\rho}_n^{a_1 a_2 \cdots a_n} \rangle_0$. We need to subtract this term from $\rho_n^{a_1\cdots a_n}$ since the regularization of density operator in the thermodynamic limit is not sufficient to make their expectation values equal zero.

When inversion symmetry is present, all terms that are odd in $k_a$ can be dropped. Moreover, translational symmetry implies that $l = 1$ and $n-l = 1$ terms vanish since $\bar{\rho}_1^a |0\rangle = 0$. This, in particular, makes $\bar{S}(\boldsymbol{k}) = \mathcal{O}\big((k\ell)^4\big)$. To this order, we have:

$$\bar{S}(\boldsymbol{k}) = \frac{1}{N_{\text{orb}}} \left\langle \frac{1}{2} \left\{ \delta\bar{\rho}_2^{ab}, \delta\bar{\rho}_2^{cd} \right\} \right\rangle_0 k_a k_b k_c k_d \ell^4 \,. \tag{A.5}$$

$\bar{\rho}_2^{ab}$ are generators of area preserving linear transformations that leave the origin invariant, i.e. rotation, shear aligned along $xy$-axes and shear aligned at $45^0$ angle with respect to the axes. This can be seen explicitly as follows:

$$[\bar{\rho}_2^{ab}, \delta\bar{\rho}(\boldsymbol{k})] = -\frac{ik_c}{2} \left( \epsilon^{ac} \partial^b \delta\bar{\rho}(\boldsymbol{k}) + \epsilon^{bc} \partial^a \delta\bar{\rho}(\boldsymbol{k}) \right), \tag{A.6}$$

$$e^{i\alpha_{ab}\bar{\rho}_2^{ab}} \delta\bar{\rho}(\boldsymbol{k}) = \delta\bar{\rho} \left( e^{\boldsymbol{\alpha}.\boldsymbol{\epsilon}} . \boldsymbol{k} \right) e^{i\alpha_{ab}\bar{\rho}_2^{ab}} \,, \tag{A.7}$$

where $\partial^a \equiv \partial/\partial k_a$, $\alpha_{ab}$ is a symmetric real valued matrix with three independent parameters and $(\boldsymbol{\alpha}.\boldsymbol{\epsilon})_a^{\ b} = \alpha_{ac}\epsilon^{cb}$.

The quantum metric in the parameter space formed by $\alpha_{ab}$ is related to the static structure factor via Eq. (A.5). Below, we show that the Berry curvature is related to the Hall viscosity following Refs. [2, 4, 52]. To begin, we notice that $\bar{\rho}_2^{ab}$ operators form a closed Lie algebra with the following commutation relations:

$$[\bar{\rho}_2^{ab}, \bar{\rho}_2^{cd}] = -\frac{i}{2} \left( \epsilon^{ac} \bar{\rho}_2^{bd} + \epsilon^{ad} \bar{\rho}_2^{bc} + \epsilon^{bc} \bar{\rho}_2^{ad} + \epsilon^{bd} \bar{\rho}_2^{ac} \right). \tag{A.8}$$

Under the area preserving transformation, the deformed many-body ground state becomes $|\alpha\rangle = e^{i\alpha_{ab}\delta\bar{\rho}_2^{ab}} |0\rangle$. The Berry curvature with respect to $\alpha_{ab}$ at $\alpha_{ab} = 0$ is given by:

$$\bar{\mathcal{F}}^{ab,cd} = i \left\langle [\bar{\rho}_2^{ab}, \bar{\rho}_2^{cd}] \right\rangle_0 \,. \tag{A.9}$$

Eq. A.7 corresponds to the area-preserving strain $u_b^a = -\alpha_{bc}\epsilon^{ca}$. As shown in Ref. [4], the Berry curvature with respect to strain gives the anti-symmetric viscosity tensor $\bar{\eta}_{b\ d}^{a\ c}$, where stress and the rate of strain are related via $\bar{\sigma}_b^a = -\bar{\eta}_{b\ d}^{a\ c}\partial_c v^d$ and $v^d$ is the local velocity of the fluid. We have:

$$\bar{\eta}_{b\ d}^{a\ c} = -\frac{\hbar}{A} \epsilon_{be}\epsilon_{df} \bar{\mathcal{F}}^{ae,cf} \,, \tag{A.10}$$

where $A = 2\pi\ell^2 N_{\text{orb}}$ is area of the sample.

Using Eqns. (A.8) and (A.9), we can obtain the expression for Hall viscosity. To this end, let's define the following ground state expectation value [52]:

$$\bar{\eta}_H^{ab} = -\frac{\hbar}{2\pi\ell^2}\frac{\left\langle \bar{\rho}_2^{ab}\right\rangle_0}{N_{\text{orb}}}\,. \tag{A.11}$$

The expressions for $\bar{\eta}_H^{ab}$ and $\bar{\eta}_{b\ d}^{a\ c}$ simplify in the presence of rotational symmetry. In this case, the generator of rotation, i.e., the angular momentum $\hbar g_{ab}\bar{\rho}_2^{ab} = \Delta\bar{L}$ is conserved. As such, we can write:

$$\bar{\eta}_H^{ab} = -\frac{g^{ab}}{4\pi\ell^2}\frac{\Delta\bar{L}}{N_{\text{orb}}}\,. \tag{A.12}$$

$\Delta\bar{L}$ is the regularized (not by subtracting the ground state value but $\nu$ times the value for a fully filled LL) angular momentum on the plane. To obtain an expression in terms of guiding center shift, we note that on a sphere, the angular momentum of an electron is: $\bar{L}^j/\hbar \in \left\{-\frac{N_{\text{orb}}-1}{2}, -\frac{N_{\text{orb}}-1}{2}+1, \cdots, \frac{N_{\text{orb}}-1}{2}\right\}$ with $\sum_j \bar{L}^j = 0$ since the ground state is inversion symmetric. On the other hand, the angular momentum operator on the plane is of the form $\bar{L}^j/\hbar = g_{ab}\bar{R}_j^a\bar{R}_j^b/2\ell^2 = m + 1/2$, with $m \in \{0,1,2,\cdots\}$ indexing the orbitals. We can convert the former to latter by shifting the angular momentum of each orbital by $\hbar N_{\text{orb}}/2$. This gives the unregularized angular momentum on the plane to be $\bar{L}/\hbar = N_e N_{\text{orb}}/2$. To obtain its regularized counterpart $\Delta\bar{L}$, we regularize the density operator by placing charge $-\nu$ on each orbital with index $m \in \left\{0, 1, \cdots, \nu^{-1}N_e - 1\right\}$. Thus, we obtain:

$$\begin{aligned}\frac{\Delta\bar{L}}{\hbar} &= \frac{\bar{L}}{\hbar} - \frac{\nu(\nu^{-1}N_e)^2}{2}\\ &= -\frac{N_e\bar{\mathcal{S}}}{2} = N_{\text{cb}}S_{\text{GC}}\,.\end{aligned} \tag{A.13}$$

Here $N_{\text{cb}} = N_e/p$ is the number of composite-bosons, and $\bar{\mathcal{S}} \equiv \nu^{-1}N_e - N_{\text{orb}}$ is the guiding center shift quantum number. The appropriately symmetrized Hall viscosity tensor with all indices lowered becomes:

$$\bar{\eta}_H^{ab} = -\frac{\hbar}{8\pi\ell^2}\frac{2S_{\text{GC}}}{q}g^{ab}\,, \tag{A.14}$$

$$\bar{\eta}_{ab,cd} = -\frac{\hbar}{16\pi\ell^2}\frac{2S_{\text{GC}}}{q}\left(g_{ac}\epsilon_{bd} + g_{ad}\epsilon_{bc} + g_{bc}\epsilon_{ad} + g_{bd}\epsilon_{ac}\right)\,. \tag{A.15}$$

The guiding center Hall viscosity can be read off $\bar{\eta}_H = \bar{\eta}_{12,22}/g_{22} = -\frac{\hbar}{8\pi\ell^2}\frac{2S_{\text{GC}}}{q}$.

We now derive the Haldane bound for the general case without rotational symmetry. To this end, we note that if we define the state deformed by area preserving transformation as: $|\alpha\rangle = e^{i\alpha_{ab}\delta\bar{\rho}_2^{ab}}$, then the quantum metric $\bar{\mathcal{G}}^{ab,cd}$ is related to the static structure factor as follows:

$$\bar{\mathcal{G}}^{ab,cd} = \frac{1}{2}\left\{\delta\bar{\rho}_2^{ab}, \delta\bar{\rho}_2^{cd}\right\}\,, \tag{A.16}$$

$$\bar{S}(\boldsymbol{k}) = \frac{\bar{\mathcal{G}}^{ab,cd}}{N_{\text{orb}}}k_a k_b k_c k_d \ell^4\,. \tag{A.17}$$

The general inequality between the quantum metric and Berry curvature, i.e. $\bar{\mathcal{G}}^{ab,cd} \pm i\bar{\mathcal{F}}^{ab,cd}/2$ are positive semi-definite, gives the Haldane bound. The inequality is derived in subsection A.1.

The expression of Haldane bound simplifies in the presence of rotational invariance. We can write a general expression of the quantum metric tensor assuming rotational symmetry as follows:

$$\frac{\bar{\mathcal{G}}^{ab,cd}}{N_{\text{orb}}} = \bar{\kappa}\left(g^{ac}g^{bd} + g^{ad}g^{bc}\right) + (\bar{S}_4 - 2\bar{\kappa})g^{ab}g^{cd} + (\bar{\kappa} - \bar{S}_4)\left(\epsilon^{ac}\epsilon^{bd} + \epsilon^{ad}\epsilon^{bc}\right), \quad \text{(A.18)}$$

where $\bar{S}(\boldsymbol{k}) = \bar{S}_4\left(g^{ab}k_ak_b\ell^2\right)^2$ and the coefficients in the above expression are fixed by the conservation of angular momentum, i.e. $\bar{\mathcal{G}}^{ab,cd}g_{cd} = 0$. Using Eq. (A.14), we obtain the Haldane bound for the isotropic case:

$$\bar{S}_4 \geq \frac{\pi\ell^2}{\hbar}|\bar{\eta}_H|. \quad \text{(A.19)}$$

## A.1 Inequality between quantum metric and Berry curvature

In this subsection we provide a derivation of the inequality between the quantum metric and Berry curvature. To this end, let's assume that upon varying the real parameters $x^\mu$, the ground state transforms to $|\Psi\rangle = e^{iO_\mu x^\mu}|0\rangle$ with $\mu = 1, 2, \cdots n$. We take $O_\mu$ to be hermitian operators with their ground state expectation values subtracted off, i.e. $\langle 0|O_\mu|0\rangle = 0$. The quantum metric $\mathcal{G}_{\mu\nu}$ and Berry curvature $\mathcal{F}_{\mu\nu}$ are given by:

$$\begin{aligned}
\mathcal{G}_{\mu\nu} &= \text{Re}\left[\langle\partial_\mu\Psi|(1 - |\Psi\rangle\langle\Psi|)|\partial_\nu\Psi\rangle\right] \\
&= \frac{1}{2}\left\langle\{O_\mu, O_\nu\}\right\rangle_0, \quad \text{(A.20)} \\
\mathcal{F}_{\mu\nu} &= -2\,\text{Im}\left[\langle\partial_\mu\Psi|\partial_\nu\Psi\rangle\right] \\
&= i\left\langle[O_\mu, O_\nu]\right\rangle_0. \quad \text{(A.21)}
\end{aligned}$$

Now, let's construct the quantum geometric tensors $\mathcal{T}^\pm_{\mu\nu} = \mathcal{G}_{\mu\nu} \pm i\mathcal{F}_{\mu\nu}/2$ with:

$$\mathcal{T}^-_{\mu\nu} = \left\langle O_\mu O_\nu\right\rangle_0. \quad \text{(A.22)}$$

We have $\mathcal{T}^-_{\mu\nu}\alpha^{\mu*}\alpha^\nu = \left\langle(\alpha^\mu O_\mu)^\dagger(\alpha^\nu O_\nu)\right\rangle_0 \geq 0$, i.e. $\mathcal{T}^-_{\mu\nu}$ is a positive semi-definite matrix. Similar inequality can be derived for $\mathcal{T}^+_{\mu\nu}$. Another way of expressing the inequality is $\mathcal{G}_{\mu\nu}\alpha^{\mu*}\alpha^\nu \geq \frac{1}{2}\left|\mathcal{F}_{\mu\nu}\alpha^{\mu*}\alpha^\nu\right|$.

# B Conditions for saturation of the Haldane bound

To obtain the conditions for the saturation of the bound, let us split $\bar{\rho}_2^{ab}$ into the following three components:

$$\bar{\rho}_2^{ab}k_ak_b = \bar{\rho}_2^{-2}(k_+)^2 + \bar{\rho}_2^0 k_+k_- + \bar{\rho}_2^2(k_-)^2, \quad \text{(B.1)}$$

$$g_{ab} = \delta_{\alpha\beta}e_\alpha^a e_\beta^b, \quad g^{ab} = \delta^{\alpha\beta}E_\alpha^a E_\beta^b, \quad e_\alpha^a E_b^\alpha = \delta_b^a, \quad \text{(B.2)}$$

$$k_\pm = E_x^a k_a \pm iE_y^b k_b. \quad \text{(B.3)}$$

We have:

$$\bar{\rho}_2^0 = \frac{g_{ab}\bar{\rho}_2^{ab}}{2} = \frac{\Delta\bar{L}}{2\hbar}, \quad \text{(B.4)}$$

$$\bar{\rho}_2^{\pm 2} = \frac{\rho_2^{ab}e_a^\alpha e_b^\beta M^\pm_{\alpha\beta}}{4}, \quad \text{(B.5)}$$

$$M^\pm_{\alpha\beta} = \sigma^z_{\alpha\beta} \pm i\sigma^x_{\alpha\beta}, \quad \text{(B.6)}$$

where $\sigma^j$ are Pauli matrices. The non-trivial commutation relations are:

$$\left[\bar{\rho}_2^0, \bar{\rho}_2^{\pm 2}\right] = \pm \bar{\rho}_2^{\pm 2}, \tag{B.7}$$

$$\left[\bar{\rho}_2^2, \bar{\rho}_2^{-2}\right] = -\frac{\bar{\rho}_2^0}{2}, \tag{B.8}$$

For Haldane bound to be saturated, one of the two linear combinations $\mathcal{T}_\pm^{ab,cd} \equiv \bar{\mathcal{G}}^{ab,cd} \pm i \bar{\mathcal{F}}^{ab,cd}/2$ must have identically zero components. This leads to the following two sets of general conditions:

$$\langle \delta \bar{\rho}_2^{2s} \delta \bar{\rho}_2^{2s} \rangle_0 = 0, \tag{B.9}$$
$$\text{and}$$
$$\left[\langle \delta \bar{\rho}_2^0 \delta \bar{\rho}_2^2 \rangle_0 = \langle \delta \bar{\rho}_2^{-2} \delta \bar{\rho}_2^2 \rangle_0 = \langle \delta \bar{\rho}_2^{-2} \delta \bar{\rho}_0^2 \rangle_0 = 0, \right.$$
$$\text{or}$$
$$\left. \langle \delta \bar{\rho}_2^2 \delta \bar{\rho}_2^0 \rangle_0 = \langle \delta \bar{\rho}_2^2 \delta \bar{\rho}_2^{-2} \rangle_0 = \langle \delta \bar{\rho}_0^2 \delta \bar{\rho}_2^{-2} \rangle_0 = 0 \right], \tag{B.10}$$

where $s \in \{1, 0, -1\}$ and is not summed over. The $s = 0$ condition in Eq. (B.9) implies that angular momentum $\Delta \bar{L} = 2\hbar \bar{\rho}_2^0$ cannot have any fluctuations around its mean value, i.e. saturation of Haldane bound requires SO(2) rotational invariance.

With rotational invariance, the conditions can be simplified to the following:

$$\langle \bar{\rho}_2^{-2} \bar{\rho}_2^2 \rangle_0 = 0, \quad \text{or} \quad \langle \bar{\rho}_2^2 \bar{\rho}_2^{-2} \rangle_0 = 0, \tag{B.11}$$

where we have used the fact that $\delta \bar{\rho}_2^{\pm 2} = \bar{\rho}_2^{\pm 2}$. Since $\rho_2^{-2} = \left(\rho_2^2\right)^\dagger$, the LHS of both subconditions are positive semi-definite. Therefore, we require:

$$\bar{\rho}_2^{-2} |0\rangle = 0, \quad \text{or} \quad \bar{\rho}_2^2 |0\rangle = 0, \tag{B.12}$$

i.e. the spin-2 graviton excitation must be maximally chiral. On a 2D plane when the quantum Hall droplet has a finite radius, there are gapless edge excitations in addition to the gapped bulk ones. The above condition pertains to the bulk gravitons only. In the presence of edges, the condition for saturation of the bound becomes that one of the operators $\bar{\rho}_2^{\pm 2}$ must create purely an edge excitation.

## C  Interpretation of the spectral sum rules of Ref. [25]

In Ref. [25], spectral sum rules are derived for isotropic FQH states. Two of them relate the sum and difference of the spectral weights of two graviton chiralities to $\bar{S}_4$ and guiding center shift/Hall viscosity respectively. Using the expressions of $\rho_T(\omega), \bar{\rho}_T(\omega)$ given in the reference, we can define $I \equiv \nu \int_0^\infty \frac{d\omega}{\omega^2} \rho_T(\omega)$ and similarly $\bar{I}$. For the lowest Landau level FQH states, the sum rules can then be stated as:

$$I + \bar{I} = \bar{S}_4, \tag{C.1}$$

$$I - \bar{I} = \frac{\nu(\mathcal{S} - 1)}{8}. \tag{C.2}$$

Using Eqns. (A.2), (B.1), (B.4), (B.8) and (A.13), if we identify:

$$I = \frac{1}{N_{\text{orb}}} \langle \bar{\rho}_2^2 \bar{\rho}_2^{-2} \rangle_0, \tag{C.3}$$

$$\bar{I} = \frac{1}{N_{\text{orb}}} \langle \bar{\rho}_2^{-2} \bar{\rho}_2^2 \rangle_0, \tag{C.4}$$

the above spectral sum rules are reproduced by the guiding center approach presented in this paper.

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
