# Peer review of "A numerical study of bounds in the correlations of fractional quantum Hall states"

_SciPost Physics, doi:SciPost Phys. 16, 117 (2024)_

## Round 2 · Referee Report · Ajit Coimbatore Balram · 2023-5-29

Strengths
A detailed comparison between field-theoretic predictions and microscopics (numerics based on the DMRG method) has been carried out for the guiding center-static structure factor for many FQH states.
Numerical evidence is presented to show that weak maximal chirality is insufficient to saturate the Haldane bound.
An intriguing conjecture on the coefficient of (k\ell)^6 being bounded by the field-theoretic predictions is presented which will hopefully spurn interest in analytically investigating this conjecture.
Weaknesses
Methods are restricted to states that either have a matrix product state (MPS) representation or is obtained as the ground state of a model Hamiltonian. In particular, the 2/5 Jain state is not exactly the ground state of the V1 Hamiltonian in the lowest Landau level (a point that should be clarified in the text).
The evaluations are done in the cylindrical geometry (suitable for MPS) with a finite circumference of the cylinder so the results are in the quasi-2D and not strict 2D limit.
Report
The article presents the first detailed computational evaluation using the density matrix renormalization group technique of the guiding center static structure factor of many fractional quantum Hall states. Numerical evidence has been presented to show that ideal conformal block wave functions saturate the Haldane bound while states that cannot be represented as conformal blocks do not do so. Based on topological quantum field theories, the leading coefficients of S(k) have been predicted and the current work suggests that the (k\ell)^6 coefficient is bounded above by these field-theoretic predictions. I recommend the publication of the paper in SciPost after the authors have had a chance to look at a couple of my comments given below.
One interesting case that could be worth discussing is that of the 2/3 Jain state which I suspect does not saturate the Haldane bound. Although it is not representable as an ideal conformal block wave function, it is known to have near unit overlap with the particle-hole conjugate of the 1/3 Laughlin wave function for accessible systems. The particle-hole conjugate of 1/3 Laughlin saturates the Haldane bound since its entanglement spectrum is a mirror image (left movers become right movers and vice-versa) of the 1/3 Laughlin's. The 2/3 Jain state will still satisfy the weaker condition of maximal chirality (low-lying entanglement levels would disperse in only one direction) so that it is in the same topological phase as the hole-conjugate of 1/3 Laughlin.
Can the neutral excitations be used to probe the gapped/gaplessness of a quantum Hall state? For example, can one compute the S(k) of the Gaffnian or the PH-Pfaffian and show that it represents a gapless state?
Fix minor typos given in below in ``Requested changes."
Requested changes
Along with Ref. [13] cite Phys. Rev. X 12, 021008 (2022) which deals with similar ideas.
Minor typos
1) in the organization paragraph replace ``fractional quantum Hall ground states" with ``fractional quantum Hall ground states are presented in Sec. V"
2) on page 3, first paragraph: the phrase ``particles in the" is repeated
3) in Sec. III ``behave like a bosons" --> ``behave like bosons"
Author: Prashant Kumar on 2024-03-05 [id 4342]
(in reply to Report 1 by Ajit Coimbatore Balram on 2023-05-29)Please see the attached response.
Attachment:
Referee_report_long_wavelength_static_structure_factor_UpGShBL.pdf
Author: Prashant Kumar on 2024-03-05 [id 4340]
(in reply to Report 3 on 2023-08-01)Please see the attached response.
Attachment:
Referee_report_long_wavelength_static_structure_factor.pdf

---

## Round 2 · Referee Report · Anonymous · 2023-7-19

Strengths
1- convincing numerical test of conjectured bounds of some fractional quantum Hall states
2- states of different fractional quantum Hall classes tested (Laughlin, composite-fermion and non-Abelian states)
Weaknesses
1-manuscript lacks equilibrium between previously known and new parts
2-background information provided in the most general form although only parts of it is required -> lack of readability
3-lack of detailed (physical) discussion of numerical findings
Report
The authors present a numerical study of the coefficients of the static structure factor associated with some model fractional quantum Hall (FQH) states in a low-wave-vector expansion. Due to the incompressibility of FQH states, the static structure factor is known to start with a quartic contribution, and the associated coefficient S_4 is known to be delimited by a lower bound, called the Haldane bound. Furthermore, it has been argued theoretically that the coefficient S_6 of the sixth-order term in the wave-vector expansion should encode information about the central charge in the associated edge theory. The aim of the manuscript is a numerical test (iDMRG) of these bounds, which are found to be saturated in specific model interactions (based on an intelligent choice of the pseudopotentials) for which the Laughlin and Moore-Read states are known to be the (densest) exact ground states. Furthermore, the evolution of the expansion coefficients S_4 and S_6 upon variation of the interaction potential is studied, and it is found that S_4 increases as expected while S_6 is mostly an upper bound. The only exception studied is the nu=2/5 Jain state which, for an interaction consisting only of a positive V_1 pseudopotential, has a higher value of S_6 than that predicted theoretically.
I find the results of the manuscript convincing. They provide relevant insight into the physical characterization of FQH states and thus merit publication in SciPost. However, there are several parts of the manuscript that are quite hard to follow, due to notational issues and lack of clear motivation so that I would invite the auhors to modify the respective parts in line of the detailed criticism that may be found below. I am also lacking a more physical discussion of the results, e.g. can the authors provide a physical picture or a hypothesis why the theoretical coefficient S_6 is an upper bound for the Laughlin and Moore-Read states while it seems a lower bound for the Jain states, at least the nu=2/5 state discussed here? Is that due to the projector to the lowest Landau level that needs to be taken into account in the Jain states?
Detailed criticism:
1) The authors discuss the expansion of the static structure factor in very general terms such as to obtain a leading coefficient that is a rank-4 tensor. Because eventually only isotropic cases are discussed, the reader may wonder (and actually does so) why a complicated general discussion has been chosen instead of a simplified one. There is thus some redundant information, e.g. when the symmetry of the rank-4 tensor is discussed. If the authors decide to keep the discussion on the most general level, they may do so, but they should add a sentance at the beginning pointing that out and indicating that finally they only use an isotropic version of it. This is also the case for the Euclidean metric tensor g^{ab}, which eventually boils down to the simple Kronecker symbol because no geometric difformations or anisotropies are taken into account [in Eqs. (14), above Eq. (15), Eq. (30), which by the way should be corrected since the second k_a in the Gaussian exponent should be a k_b].
2) In the same vein, the reader is confronted with a general discussion of the composite-boson picture of the FQH effect. However, only Laughlin states and composite-fermion (Jain) states are investigated numerically, along with the Moore-Read state. It might be useful to directly orient the discussion of Sec. III to composite fermions.
Minor criticism:
i) Some fundamental quantities are undefined. As an example, $\ell$ is the fundamental length scale, used since the abstract, but at no moment it is said that it is the magnetic length. Please also say explicitly that $N$ is the number of particles [below Eq. (1)]. The commutator of Eq. (5) yields precisely the (square of the) magnetic length, and this point should be mentioned explicitly.
ii) When discussing the variance below Eq. (26), the vectors t=(1,1,...,1) and s are not defined.
In conclusion, I find the manuscript interesting and will recommend publication after the authors have taken into account the above criticism. The authors should better equilibrate their work. Their main findings (the numerical tests) are presented in Sec. V, while the other sections rephrase previous work. It is indeed helpful to have a selfcontained manuscript, but there seems to be some redundant information due to the above-mentioned aim at generality in the introductory sections, information that is eventually not needed to understand the numerical results. I would therefore recommend to simplify these parts and, on the contrary, to invest a bit more Sec. V where the main results are simply described. Here, the reader is missing a bit of discussion and interpretation, as I have mentioned above. The authors may want to discuss the deviations from the bounds. Is it due to the lowest-Landau-level projection in the composite-fermion case? Why is the descrepancy in the range of several percent for the Laughlin states when the Coulomb interaction is taken into account, while exact diagonalization seems to indicate that the ground state has more than 99% of overlap with the Laughlin state at nu=1/3 and 1/5? There is a bit of place for such a discussion.
Requested changes
see report
Author: Prashant Kumar on 2024-03-05 [id 4341]
(in reply to Report 2 on 2023-07-19)Please see the attached response.
Attachment:
Referee_report_long_wavelength_static_structure_factor_LQuLOOP.pdf

---

## Round 2 · Referee Report · Anonymous · 2023-8-1

Strengths
1- Introduction, motivation, and background are well written.
Weaknesses
1- There's a lack of discussion regarding the experimental significance of the static structure factor, in particular how the coefficients may be measured.
Report
The authors examines S4 and S6 coefficients of the static structure factor, comparing them to various calculations on conformal block wavefunctions.
Using quantum Hall DMRG (or MPS construction for model wavefunctions), the authors verifies the Haldane bound on the quartic coefficient.
Further, the authors confirm the theoretical prediction for the sixth order for conformal blocks wavefunctions, and provides evidence for a general inequality.
This paper is numerically driven. It appears that all the theoretical results have previously existed in literature. It is not completely clear if Appendix A1 & B are new, or just rewriting [2]. As a numerical work, the results are clear and supports the conclusion. These results are worthy of publication in SciPost, although there can be much improvement in the manuscript.
(See requested changes)
Requested changes
1- Being a numerical-driven paper using some DMRG package, there is a notable absent of discussion on the simulation parameters and their convergence criteria. Other numerical physics in the field should be able to reproduce the data.
Regarding the Gaussian envelope
2- V(r) is missing 1/r factor (top of pg 7)
3- The Gaussian envelope leads to a order O(ell^2/xi^2) correction to many observables (e.g. ref 14), and it is natural to assume that this correction also appears in the structure factors. A proper error analysis from the effects of the Gaussian envelope should be performed. What is the convergence of the structure factors in terms of 1/xi?
4- The 2/5 Jain state admits a "model wavefunction" within the lowest and first LL (with projected delta-interaction). Maybe this model wavefunction is the closest one may get to a "maximal chiral" wavefunction. How do S4 and S6 compare to theory then?
After reading the other referee's report:
5- I also found the section on "composite bosons" confusing. This is the section where c-, s and various quantities are defined, but these quantities exist beyond the composite boson picture. The authors should try to define these topological quantities in a more general context, and then discuss their properties in the Chern-Simons case.
6- A separate analysis on the 2/3 state is of course unnecessary, seeing it is just PH of the 1/3 state. However, the behavior of all the quantities under PH is difficult to find in the current manuscript. Perhaps the authors can organize this information in a table.

---

## Round 4 · Referee Report · Anonymous (Referee 2) · 2024-4-2

Report
The authors have revised their manuscript according to my criticism in a globally satisfactory manner and, as far as I can see, also to that of the other referees. I recommend publication, but I would invite the authors to consider two minor (technical) points prior to publication.
1) Reference [24] does not contain an arXiv number. If it has already been posted on the arXiv, please add the reference, or brand it as "unpublished" otherwise.
2) In the paragraph below Eq. (5), the quantum numbers $\alpha$ and $\beta$ in the quantum states, presumably associated with the guiding-center operators, are undefined. Please add a definition so that the reader may appreciate better the argument developed in the paragraph.
plus a little typo:
In the last paragraph of Sec. III, second sentence, the authors may correct to "degrees of freedom" and not "off".

Author: Prashant Kumar on 2024-05-01 [id 4460]
(in reply to Report 1 on 2024-04-02)Dear Referee,
We express our sincere gratitude for your review of our revised manuscript, as well as for your insightful suggestions and endorsement for publication. Your recommendations have been incorporated into the final published version.
Sincerely,
Prashant Kumar, F. D. M. Haldane

---

## Round 4 · List of Changes

-
We have clarified throughout the manuscript that rotational invariance is not a requirement for the existence of Haldane bound but it is a necessary condition for the saturation. Therefore, the preliminary sections discuss the properties of static structure factor without the assumption of rotational invariance. In the later sections, where we investigate the conditions for saturation of the bound, we specialize to the rotationally invariant case.
-
We have added discussion motivating the usage of composite-boson formulation in the manuscript.
-
Behavior of various physical quantities under Particle-Hole symmetry has been emphasized in section IV.
-
In the discussion section, we have commented on the experimental relevance of our numerical calculations. In particular, by utilizing the spectral sum rules of Golkar \textit{et al}, we can predict the relative spectral weights of spin $+2$ and $-2$ graviton excitations of FQH states. We demonstrate this via an explicit prediction for $\nu=1/3$. Further, we have added more discussions of the results obtained in section V as suggested by the referees.
-
Appendix C has been added reinterpreting the spectral sum rules of Golkar \textit{et al} in the guiding center formulation adopted in the manuscript.

---

## Editorial Decision

published